# Lateral-Type Spin-Photonics Devices: Development and Applications

**DOI:** 10.3390/mi12060644

**Published:** 2021-05-31

**Authors:** Nozomi Nishizawa, Hiro Munekata

**Affiliations:** Laboratory for Future Interdisciplinary Research and Technology, Tokyo Institute of Technology, Yokohama 226-8503, Japan; munekata.h.aa@m.titech.ac.jp

**Keywords:** spintronics, spin-photonic devices, spin-LED, spin-photodiode, circularly polarized light, spin injection, cancer detection

## Abstract

Spin-photonic devices, represented by spin-polarized light emitting diodes and spin-polarized photodiodes, have great potential for practical use in circularly polarized light (CPL) applications. Focusing on the lateral-type spin-photonic devices that can exchange CPL through their side facets, this review describes their functions in practical CPL applications in terms of: (1) Compactness and integrability, (2) stand-alone (monolithic) nature, (3) room temperature operation, (4) emission with high circular polarization, (5) polarization controllability, and (6) CPL detection. Furthermore, it introduces proposed CPL applications in a wide variety of fields and describes the application of these devices in biological diagnosis using CPL scattering. Finally, it discusses the current state of spin-photonic devices and their applications and future prospects.

## 1. Introduction

### 1.1. Spin-Polarized Light Emitting Diodes (Spin-LEDs)

In semiconductor spintronics research, whose purpose is to employ the spin degree of freedom in semiconductors, injecting spin-polarized carriers from ferromagnetic materials into semiconductors has been a central topic since the 1990s. Research on spin injection was motivated by the spin-polarized field-effect transistors (spin-FETs) proposed by Datta and Das [1]. The mechanism underlying spin-FETs can be seen as an analogy of the giant magnetoresistance (GMR) effect [2,3] for semiconductors, in which the inversion of the ferromagnetic layer for GMR corresponds to the spin rotation of conduction electrons using the gate electric field in spin-FETs. The realization of the suggested spin-FETs requires electrical injection, control, and detection of spin-polarized carriers. In the beginning, spin injection and detection were investigated simultaneously; however, only a few credible experimental results were obtained, indicating the need for investigating spin injection and detection independently. In 1999, Fiederling et al. [4] and Ohno et al. [5] almost simultaneously reported spin injection from magnetic semiconductors into non-magnetic semiconductor heterojunctions and the resulting emission of circularly polarized light (CPL). Fiederling et al. achieved electron spin injection from the II–VI magnetic semiconductor BeZnMnSe into an electrically pumped GaAs/AlGaAs light-emitting diode (LED) in the Faraday configuration, in which the spin injection efficiency was approximately 86%. The paramagnetic semiconductor BeZnMnSe exhibits a large Zeeman splitting under strong magnetic fields of several Tesla at extremely low temperatures (1.5–33 K), leading to this large injection efficiency. In contrast, Ohno et al. achieved hole spin injection from the ferromagnetic semiconductor GaMnAs in the remnant state into an InGaAs quantum well (QW) heterostructure. The magnetization of the GaMnAs layer in the remnant state pointed toward the in-plane easy axis, and CPL could be obtained in the same direction as the magnetization. The magnetic field dependence of the obtained CPL traced the hysteresis loop of the magnetization of the GaMnAs electrode, although neither photoluminescence from the same device nor electroluminescence (EL) from the QW structure without GaMnAs showed such a hysteresis. It has been reported that these results are attributable to the hole spin injection. Since these reports, experiments involving CPL emission measurement from various combinations of magnetic materials and semiconductor heterostructures have been performed to demonstrate the injection of spin-polarized carriers. Such devices consisting of ferromagnetic electrodes on semiconductor LED structures are called “spin-polarized light emitting diodes (spin-LEDs)” [6,7,8]. 

In a spin-LED, spin-polarized carriers are injected from a ferromagnetic electrode into a forward biased semiconductor toward a *p-i-n* junction. The injected spins travel via drift and diffusion across a spacer layer (an upper cladding layer) and reach the active region, typically comprising single or multiple QWs, quantum dot layers, or thick well layers. The surviving spin-polarized carriers undergo radiative recombination with the unpolarized counterparts according to the optical selection rules. In the case of GaAs, which is a direct bandgap semiconductor with a thick active layer, the valence bands mainly consist of two-fold degenerate *P*_3/2_ heavy-hole and light-hole subbands at Γ_8_, whereas the conduction band is a two-fold degenerate *S*_1/2_ state at Γ_6_. According to the optical selection rules [9], two transition processes from *S*_1/2_ to *P*_3/2_ are allowed. The heavy-hole subband transition probabilities are three times larger than those of the light-hole subband, leading to light emission with a maximum degree of circular polarization (DOCP) of 50% for an injection of fully (100%) polarized electrons. In contrast, the degeneracy of heavy- and light-hole subbands are resolved in QW structures; the maximum DOCP is 100% because only the heavy-hole subband transitions predominantly occur in this case. In all cases, the spin angular momentum of the carriers is transferred to the emitted light through radiative recombination, resulting in the emission of light with angular momentum, that is, CPL. The observed DOCPs are directly associated with the spin polarization of the injected carriers, which are utilized to evaluate the spin polarization in magnetic electrodes and investigate the spin dynamics in semiconductors.

In the earliest studies on spin-LEDs [5,10,11,12], diluted ferromagnetic semiconductors (DMSs) were used as spin-source electrodes. However, most of the DMSs exhibit low Curie temperatures, placing an upper limit on the spin-LED operation at temperatures lower than room temperature (RT). Therefore, instead of DMSs, the collective focus shifted toward the study of spin-LEDs with ferromagnetic metals; however, an unavoidable problem emerges in spin-injection devices with metal/semiconductor junctions. Spin injection from a ferromagnetic metal into a semiconductor via an Ohmic junction is difficult in principle because of an inherent limitation of the electron diffusion process. This limitation exists because of the large difference in the electrical conductivity between these materials, the so-called “conductivity mismatch” [13,14,15,16]. To circumvent this obstacle, the introduction of an additional tunneling barrier between the ferromagnet and semiconductor was proposed [14,15]. Based on this proposal, various junctions for spin injection have been investigated. First, the spin injection was performed via the Schottky barrier as a triangular tunneling barrier. Zhu et al. [17] reported CPL emission with a DOCP of 2% on a spin-LED comprising an Fe electrode and InGaAs QW at 300 K. This observed emission can be explained using the concept of tunneling through the Fe/GaAs Schottky barrier. Subsequently, Hanbicki et al. [18] reported a large DOCP of 4% at 240 K on a spin-LED consisting of an Fe and a GaAs QW with a well-engineered Fe/Al_0.1_Ga_0.9_As Schottky barrier. Instead of an Fe layer, MnSb [6,19,20], NiMnSb [6], Co_2_MnGe [21], Co_2.4_Mn_1.6_Ga [22], and MnAs [23] have been investigated as the spin sources on GaAs-based Schottky barriers. Although spin injection has been achieved in these studies, the resulting DOCPs of the emitted CPL have been approximately 30% at low temperatures and a few percent at RT. One possible reason for the low polarization is the interdiffusion between the magnetic metal and the semiconductor. The other is the pinning of the surface Fermi level within the gap, which may disturb the electron injection [6]. 

These issues are absent from the ferromagnet–insulator–semiconductor (MIS) structure, where tunneling injection through an ultrathin insulator layer overcomes the conductivity mismatch. In this case, an amorphous Al_2_O_3_ tunnel barrier is used as the tunneling barrier. Motsnyi et al. [24] reported a DOCP of 2.2% at 80 K in a CoFe/AlO*_x_*/AlGaAs/GaAs spin-LED under an oblique magnetic field of several Tesla. Subsequently, the same group has continuously studied this structure and achieved DOCPs of 7% at 80 K and 1.2% at 300 K by utilizing an improved oxidation process for the AlO*_x_* layer [25]. Schottky and AlO*_x_* barriers show almost the same spin injection efficiency; however, the emission intensity from a device with an AlO*_x_* barrier is significantly higher than that with a Schottky barrier because nonradiative centers are generated by the interdiffusion at the direct metal/semiconductor junction [26,27]. Consequently, AlO*_x_* effectively suppresses the chemical reactions between the magnetic metal and semiconductor, rather than enhancing the spin injection efficiency. 

Moreover, MgO tunneling barriers, which have already achieved great success in a spin injection on tunneling magnetoresistance (TMR) junctions [28], have been introduced into spin-LEDs. Jiang et al. [29] reported that a CoFe/MgO(100) spin injector with a GaAs-based QW spin-LED exhibited EL with DOCPs of 57% at 100 K and 47% at 290 K by applying a large magnetic field of 5 T. Salis et al. [30] found a CoFe/MgO(100) spin injector with a temperature-independent spin injection efficiency of ~70% from 10 K to RT. At the time of writing this review, this spin injection efficiency is the highest reported at RT. Manago et al. [31] reported EL with a DOCP of approximately 10% on an Fe/MgO/GaAs-based LED structure at RT under 1.3 T. Furthermore, Lu et al. [32] demonstrated 32% CPL emission from a CoFeB/MgO spin injector on a GaAs-based LED at 100 K under 0.8 T. MgO tunnel barriers also achieved great success in spin-LEDs as well as on TMR junctions, However, the crystallization of MgO films requires a high substrate temperature to obtain a highly oriented film on the surface of GaAs (001), and delicate control is necessary for the interface preparation because of the large lattice mismatch between MgO and GaAs. 

Nitride-based spin-LEDs have also been studied. Similar to GaAs-based spin-LEDs, spin injection from DMS materials such as GaMnN [33], MnZnO [34], GaCrN [35], and GaN:Gd [36], as well as Al_2_O_3_ [37,38] and MgO [39,40], has been reported. In the first decade of research in this field, no unambiguous demonstration of spin injection surfaced despite the weak spin–orbit coupling and resulting long spin relaxation time in GaN. However, CPL emission with DOCPs of approximately 10% has been successfully reported at RT [36,37,38].

### 1.2. Spin-LEDs as a Circularly Polarized Light Source

As mentioned earlier, most studies on spin-LEDs have been conducted with a recognition of a spin-LED merely as a laboratory tool to investigate spin-injection and spin-dependent phenomena. A few studies have been conducted from the perspective of CPL source devices. The light emitted by a conventional light source can be converted into CPL through a linear polarizer and a quarter-wave plate (QWP). The CPL thus obtained has a high polarization. However, massive optical components, for example, a setup on an optical bench in the laboratory, are necessary and polarization control requires mechanical rotation of the QWP. Moreover, periodic polarization oscillations of CPL can be obtained by passing it through a photoelastic modulator instead of a QWP [41]; however, such devices may only be available in laboratories. A compact device that can emit CPL directly is required for practical applications. Therefore, various CPL emitters have been studied, including organic light-emitting diodes (OLEDs) with chiral polymers [42], chiral photonic crystal (metamaterial) devices [43,44], chiral light-emitting transistors [45], spin-optoelectronic devices based on hybrid organic–inorganic perovskites [46,47], spin-polarized vertical-cavity surface-emitting lasers (spin-VCSELs) [48,49,50,51,52], and spin-LEDs. All of these devices have various advantages and disadvantages for practical use in terms of polarization, intensity, and polarization controllability. An OLED with chiral polymers and chiral photonic crystals can emit CPL with a high DOCP and high intensity, whereas the polarity results from the constituent materials and structure. Using metamaterials with Archimedean spiral shapes, enantiomeric switching has been demonstrated by selecting the deformation direction by pneumatic force, whose switching speed is of the order of kilohertz. In addition, the electrical controllability of polarization has been demonstrated in electric-double-layer transistors with monolayers of transition metal dichalcogenides [45,53]. However, polarization switching is accompanied by a shift in the emission wavelength, which is an outstanding issue hindering practical applications. Although optically pumped spin-VCSELs have achieved high DOCPs with high intensity and high-speed modulation, the requirement of an additional excitation light source narrows their application range. 

Spin-LEDs are also among the candidates; however, they were once considered unsuitable for practical use as light sources because of their relatively low DOCPs at RT and the essential requirement of an applied high magnetic fields. However, the great potential of spin LEDs for use in practical CPL applications was recently demonstrated as they satisfy the following requirements for CPL sources: (1) Compactness and integrability, (2) stand-alone (monolithic) nature, (3) RT operation, (4) emission with high DOCPs, (5) polarization controllability, and (6) CPL detection (Figure 1). Satisfaction with the first requirement is essential for practical applications. Equipment set on an optical bench has limited usability in laboratories or factories. The second condition indicates that the device should be operable without the aid of another device or high power consumption, thereby expanding the application range of CPL. The third item is the minimum requirement for general use at RT, and a higher temperature tolerance is required for use in harsh environments. For the fourth requirement, a higher polarization is preferred to maximize the effectiveness of CPL. Although the fifth and sixth items are sufficient conditions rather than necessary ones, these increase the convenience of utilization in CPL applications. Arbitrary or periodic control of circular polarization with a high speed i.e., the fifth requirement, enhances the suitability for application in information technology as well as sensitivity for utilization in sensors. CPL detection may be performed with a different device; however, if one device can serve as both an emitter and a detector of CPL, then this advantageous feature may be used in various applications, including bidirectional communication and ambient light detection.

This topical review demonstrates the potential of spin-photonic devices, including spin-LEDs and related devices, for practical uses. The first half of this review describes the functions of spin-photonic devices according to the above-mentioned requirements. In particular, it focuses on side-facet emission-type spin-LEDs, hereafter called lateral-type (LT) spin-LEDs, which are considered to be more suitable for practical CPL devices for the reasons mentioned below. The second half introduces various proposed CPL applications and then discusses the utilization of spin-LEDs in biomedical applications, which are among the realizable applications.

## 2. Compact, Integrable, and Stand-Alone Spin-LEDs

Spin-LEDs intrinsically satisfy the requirements of compactness and integrability because they are based on semiconductors. In a typical LED chip, a small electrode pad is located on a large semiconductor wafer. Carriers injected by the electrode into the semiconductor flow vertically into an active layer, with some spreading in the outward direction from the electrode. To prevent interference between the carriers injected by the adjacent electrodes, the current-spreading width should be considered, which depends on the operating current density and distance between the electrode and active layer [54,55]. The current-spreading widths are estimated to be approximately 5.5 and 1.8 μm at current densities of *J* = 10 and 100 A/cm^2^, respectively, which are typical operating current densities of spin-LEDs. Simply considered, the current-spreading width provides an integration limit for the LEDs. From a magnetic perspective, there is another integration limit for preventing magnetic interactions between the electrodes. The stray magnetic field from one electrode penetrates the other magnetic electrode, which influences the magnetization. The magnetic interaction may work further than the current-spreading width; however, the magnetic interaction can be reduced by utilizing the device arrangement. The magnetization-produced magnetic force lines form stable closed loops, similar to the behavior of the polarization-controllable spin-LEDs described in Section 4.

CPL sources used with integration should be operated without an external electromagnet or another excitation light source because these elements cause spatial constraints and restrict the energy gain. The requirement of an external magnetic field is generally thought to be the weakest aspect of spin-LEDs, making these devices less attractive than the other CPL sources. Most ferromagnetic thin films exhibit in-plane magnetic anisotropy. To extract CPL through radiative recombination in a semiconductor, carriers must be spin polarized in the direction of light emission. A vertical magnetic field larger than the anisotropy field of the electrode is required to align the spins perpendicular to the surface, resulting in CPL emission from the device. Therefore, most of the DOCPs reported for spin-LEDs, as shown in the previous section, were obtained under the high magnetic fields of a few Tesla, whereas the DOCPs under zero field is negligible. There are two solutions to obtain CPL emission without an external magnetic field: Remnant state spin injection using a ferromagnetic electrode exhibiting perpendicular magnetic anisotropy (PMA) and CPL extraction from the side facets of spin-LEDs. The PMA characteristics of metal alloys and multilayers, which have been studied extensively on magnetic recording media [56], have been applied to spin-LEDs. In spin-LEDs with MnGa [57,58], and CoPt [59,60], which exhibit PMA as a bulk characteristic, surface CPL emission with DOCPs of a few percent has been reported in the remnant state. The thin CoFeB/MgO spin injector possesses a strong PMA owing to the interfacial anisotropy at the ferromagnet/oxide interface [61], which yields DOCPs of 19% at 10 K [62], 22% at 15 K, 25% at 30 K [63,64], and 7% at 300 K [65] in the remnant state. Moreover, the interface-originating PMA in metal multilayers can be utilized for CPL emission from a spin-LED structure at zero magnetic field. Fe/Tb multilayers yield DOCPs of approximately 4.4% with RT spin injection into GaAs QWs [66], and DOCPs of approximately 2.7% with that into InAs quantum dots. A DOCP of 3% at 20 K from a spin-LED with Co/Pt multilayers has also been reported [67]. The other solution is to conduct spin injection from the remnant state of an in-plane ferromagnetic film and to extract light from the side facet of the device chip normal to the remnant magnetization direction. In this article, this type of spin-LED is defined as an LT-spin-LED, in contrast to the vertical-type (VT-) spin-LEDs that emit CPL from the surface. The details of VT-spin-LEDs can be found elsewhere; the following sections focus on the LT-spin-LEDs.

## 3. Emission with High Circular Polarization

### 3.1. Crystalline AlO_x_ Tunnel Barrier Layer on GaAs

High DOCP emission requires electrons with high spin polarization in the active layer of the semiconductor LED structure and can be realized by a highly efficient spin injection through a tunneling barrier layer. Therefore, the tunneling barrier quality significantly affects the DOCP of the emission. What factors determine the tunneling barrier quality? Generally, emphasis is placed on a small quantity of pin holes, uniform thickness, and low density of interface states *D_it_*. In addition, a preparation method that does not generate nonradiative recombination centers in semiconductors is of great importance for optical devices.

A layer of silicon oxide is formed on silicon as the gate insulator layer of metal-oxide–semiconductor devices using high-temperature thermal oxidation in an oxygen atmosphere. Such a high-temperature method cannot be simply applied to III–V semiconductors. Group V elements, such as phosphorous or arsenic, have high vapor pressures, producing numerous crystal vacancies by volatilization at high temperatures. These vacancies act as nonradiative recombination centers and increase *D_it_* at the semiconductor–insulator interface. The typical *D_it_* value of AlO*_x_* and MgO layers on GaAs is approximately 10^12–14^ cm^–2^eV^–1^ [68,69], which are much larger than the value of 10^9^ cm^–2^eV^–1^ at the SiO_2_/Si interface. Alternatively, oxide layers on III–V semiconductors are formed via low-temperature post-oxidation or low-temperature deposition of the oxide material. 

Therefore, Nishizawa et al. [70] devised a natural oxidation method for epitaxial Al films; in this approach, an epitaxial Al film grown on GaAs using molecular beam epitaxy (MBE) is exposed to dry air at RT for a long time. The lattice mismatch between the GaAs and Al single crystals is small, at approximately 1.3%, and the fcc structure of the Al layer can be epitaxially grown on the As-stabilized surface of GaAs at RT. This surface forms a uniform interface with a low defect density [71,72]. Oxidized passive films with thicknesses of 4–6 Å are formed on the surface of metallic Al exposed to the atmosphere [73]. Stated differently, the penetration length of oxygen from the surface is 4–6 Å and the lower layer is not oxidized. Therefore, the epitaxial Al layers, each with a thickness of ~5.5 Å exposed under conditions close to those of natural oxidation, would form fully oxidized AlO*_x_* films and a nonoxidized interface with GaAs (however, it was later observed that the in-plane oxygen penetration length varies and is longer than that expected in some places, leading to low yields.). The resulting AlO*_x_* thickness of the 5.5 Å Al epitaxial layer is 7.0 Å. Thicker films were formed by repeating these processes. Figure 2a,b shows cross-sectional transmission electron microscope (TEM) images of 1-nm-thick AlO*_x_* on *n*-GaAs formed by post-oxidation of Al epitaxial layers grown at substrate temperatures (*T*_sub_) of 30 °C and 80 °C, respectively. The film prepared at the lower temperature exhibits uniform periodicity throughout the AlO*_x_*/*n*-GaAs interface, whereas that prepared at the higher temperature shows local disturbance in the periodic patterns at the interface. Although the left side in Figure 2b displays periodic patterns, these are broken on the right side. The formation of an epitaxial Al/GaAs interface at a low temperature reduces the number of defects at the interface [71], whereas a high substrate temperature causes numerous defects at the interface. These interface structures are inherited even after the post-oxidation process. Figure 2c shows the dependencies of *D_it_* on *T*_sub_ and thickness of AlO*_x_* (*t*_AlO*x*_), which were extracted from the admittance spectra obtained from *C-V* measurements. The samples with *T*_sub_ = 80 °C with *t*_AlO*x*_ = 0.7–3.5 nm show *D*_it_ values reminiscent of those of amorphous AlO*_x_*/GaAs junctions. In the samples with *T*_sub_ = 30 °C, the lowest value, *D*_it_ ~ 7 × 10^11^ eV^–1^cm^–2^, was obtained at *t*_AlO*x*_ = 1.0 nm. This value is lower by two orders of magnitude than the typical values of AlO*_x_*/GaAs interfaces, whereas *D*_it_ is comparable to the typical value when *t*_AlO*x*_ = 1.4 nm. The thinner Al layer enables the penetration of oxygen into GaAs, which causes crystal disturbance in the top layer of GaAs, whereas the interface of the thicker film is inferred to be ruffled by repeating the processes more than two times. In conclusion, the optimized conditions for obtaining the lowest *D*_it_ are *T*_sub_ = 30 °C and *t*_AlO*x*_ = 1.0 nm, and the resulting films are referred to as x-AlO*_x_* hereafter. The efficiency of spin injection has been assessed optically based on the DOCP values of EL obtained from spin-LED devices. The efficiency of spin injection through x-AlO*_x_* has been estimated to be approximately 63% at 5 K under no external magnetic fields. This is the highest reported efficiency among the spin-LEDs in the remnant state [70]. The efficiency has been inferred to be high, even at RT [74].

### 3.2. Circularly Polarized Emission with x-AlO_x_ Tunneling Barrier

A particular spin-LED device designed for high DOCP emission [75] includes a GaAs-based double heterostructure (DH), an x-AlO*_x_* layer, and a rectangular Fe electrode, as shown in Figure 3a. The DH elaborately designed for this spin-LED device was grown on a *p*-GaAs (001) substrate using metal–organic vapor phase epitaxy, which provides high optical quality with a few luminescence quenching centers causing nonradiative recombination. The thickness of the upper cladding layer was designed to allow (i) the suppression of spin relaxation during the travel across the cladding layer and (ii) the decrease in optical absorption at the metal electrodes. These contrary requirements can be fulfilled by utilizing short and long cladding layer, respectively. As a compromise, a 500 nm thick *n*-Al_0.3_Ga_0.7_As cladding layer was adopted, which ensured that approximately 60% of the injected spin carriers were retained at the active layers [76], whereas 4% of the generated light intensity are optically lost due to absorption at the metal electrodes [77]. In addition, a *p*-type GaAs layer, doped with 1 × 10^18^ cm^–3^ of carbon was used as an active layer in which the radiative recombination time and spin relaxation time were estimated to be approximately 10 ns and 0.1 ns, respectively, at RT [76,78]. A 1 nm thick x-AlO*_x_* layer was prepared on the top surface of the DH using MBE. This step was followed by the fabrication of 100 nm thick, 40 μm wide Au (20 nm)/Ti (5 nm)/Fe (100 nm) spin-injector stripes on top of the tunnel barrier using a separate electron beam evaporator and standard photolithog raphy. Finally, the wafer was cleaved into 1.1 mm × 2.0 mm rectangular chips. 

Figure 3b presents the helicity-dependent EL spectra obtained at RT with three different current densities, *J* = 28, 85, and 184 A/cm^2^. These spectra have peak energies in the range of approximately 1.34–1.36 eV which is below the bandgap energy (*E*_g_) of GaAs (1.42 eV at RT). In highly doped GaAs (≥ 10^18^ cm^–3^), bandtail states are formed due to the variation in the potential induced by the random distribution of charged impurities in the lattice. The light generated in such an active layer is extracted from a side facet. With this process, the optical transitions (absorption and reemission) that are associated with the bandtail states result in output emission with a peak energy lower than *E*_g_ [79,80]. In the regions with *J* < 100 A/cm^2^, there are differences between intensities of the right-handed (*σ*^+^) and left-handed (*σ*^−^) circular EL components. The DOCPs of the emitted light *P_CP_* are defined by
$P_{CP}\equiv \left\{I\left(\sigma^{+}\right)-I\left(\sigma^{-}\right)\right\}/\left\{I\left(\sigma^{+}\right)+I\left(\sigma^{-}\right)\right\}$, where *I* (*σ*^+^) and *I* (*σ*^−^) are the intensities of the *σ*^+^ and *σ*^−^ components, respectively. *P_CP_* was estimated to be 0.05 and 0.03 at *J* = 28 and 85 A/cm^2^, respectively. Although these values appeared to be small, they were notably large for LT-spin-LEDs at RT without an external magnetic field. Moreover, when *J* was further increased, the intensity of only one side of the helicity components turns reduction, resulting in a steep increase in *P_CP_* to approximately 0.98. Figure 3c,d depicts the *J* dependences of the EL intensities of *I* (*σ*^+^) and *I* (*σ*^−^) components and the *P_CP_* data obtained from three different chips, respectively. The intensity *I* (*σ*^−^), associated with the minority spins, tends to saturate in the region *J* ~ 40–80 A/cm^2^, beyond which it decreases, whereas the intensity *I* (*σ*^+^) corresponding to the majority spin, increases linearly throughout the *J* region. Consequently, nearly pure CPL emission was achieved when *J* reached ~100 A/cm^2^ or higher. Notably, the decrease in the minor circular intensity component in the high *J* region is reversible; this observed decrease in the intensity is not a transient behavior due to chip degradation or optical setup failure. These extremely high DOCPs have also been quantitatively observed using a calibrated polarimeter. The inset of Figure 3b depicts the date with *J* = 184 A/cm^2^ on a Poincaré sphere.

Let us discuss the mechanism of DOCP enhancement. Spin-polarized electrons injected from a magnetic electrode travel in the cladding layers and reached a thick active layer. Here, the electrons undergo radiative recombination with the heavy and light holes near the degenerated Γ point, producing CPL with a *σ*^+^:*σ*^−^ ratio of 3:1 in principle [9]. Therefore, within the limit of low charge/spin injection, even if the injected electrons are fully polarized, only 50% of the polarized CPL can be obtained. In the structures of the tested devices, the spin polarization of Fe was 0.42 at best, and the 500 nm *n*-Al_0.3_Ga_0.7_As cladding layer reduced the spin polarization of traveling electrons to approximately 60%, mainly due to D’yakonov-Perel’ spin scattering at RT [76,81]. The electron spin polarization and the resulting *P_CP_* were not more than 0.25 and 0.13, respectively, which were the values when the spin injection efficiency of the x-AlO*_x_* layer was assumed to be 1.0. However, the observed *P_CP_* values were much higher than those expected and could not be explained fully by the simple and existing emission processes. These surprising experimental results indicate some kind of nonlinear effect, working as a positive feedback process, between the spin-polarized carriers and circularly polarized photons during the propagation in the waveguide-like GaAs active layer until reaching the cleaved edge. We propose some possible nonlinear effects: A spin-induced birefringence effect, spin-dependent reabsorption (a helicity-dependent Moss-Burstein effect), optical spin-axis conversions presumably assisted by phonons, or combinations of these effects. The elucidation of the mechanism based on these hypotheses requires extensive and cumulative experimental results obtained through systematic experiments. However, the x-AlO*_x_* layers in the spin-LED devices hinder the reproducibility of intermittent current densities larger than 100 A/cm^2^, which impedes the collection of experimental results irrespective of the fluctuations in the sample.

### 3.3. Oxidized Al/AlAs Tunneling Barrier

Various approaches have been tested to improve the reproducibility of spin-LEDs derived from a tunneling barrier and stabilize the device performance. Consequently, it was found that epitaxial growth of an Al/AlAs bilayer on top of a DH and subsequent natural oxidation with dry air or pure oxygen resulted in a relatively high yield in spin-LEDs (~60%) [82]. Hereafter, spin-LEDs with x-AlO*_x_* tunneling barriers are called as first-generation (1G-) spin-LEDs, whereas those with oxidized Al/AlAs tunneling barriers are called the second-generation (2G-) spin-LEDs. 

Comparing the cross-sectional TEM images (Figure 2a) obtained from 1G-spin-LEDs with that of 2G-spin-LEDs (Figure 4a) shows different crystal structures. This finding reveals that the upper layer of the resulting oxidized layer is amorphous, leaving an unoxidized AlAs layer underneath the oxide layer. Before oxidation, Al and AlAs epitaxial layers have thicknesses of 0.7 nm and 2.0 nm, respectively. After oxidation and metal deposition, the oxidized amorphous layer and unoxidized layer have thicknesses of approximately 1.7 nm and 1.3 nm, respectively. The TEM images obtained from the 2G-spin-LEDs indicate that the inserted AlAs layer plays an important role as a guard for GaAs, and the oxidation progresses into the Al as well as the AlAs layers; however, it is completely interrupted by the AlAs layer over a large area, irrespective of the variation in the oxygen penetration length. The addition of an insulating AlAs layer increases the overall resistance of the device and provides robust electrical characteristics. The effects of the oxidized Al/AlAs bilayer on the electrical properties can also be deduced in terms of energy band alignment [82,83], and voltage distribution. The topmost Al layer and most of the AlAs layer is oxidized, resulting in the formation of a tunnel barrier with a height sufficient for both electrons and holes. Notably, the conduction and valence band edges of a residual underneath the AlAs layer at the Γ point are above and below those of the *n*-Al_0.3_Ga_0.7_As spin transport layer, respectively. This characteristic suggests that the overall thickness and barrier height of the tunnel barrier increase for both carriers in comparison with those of the x-AlO*_x_* tunneling barrier alone. Consequently, the oxidized Al/AlAs bilayer reduces the unwanted current paths caused by metallic pinholes, parasitic hole current, and the current concentration underneath the stripe Fe electrode. In addition, the substantial potential barrier for holes in the valence band stabilizes the hole accumulation near the oxide–semiconductor interface [32,84,85]. This increases the distribution ratio of the voltage applied to both sides of the tunneling barrier against the voltage applied to the *p-n* junction. This effect results in an efficient spin injection, even at low voltages.

The contributions of AlAs insertion to the electrical properties influence the CPL emission characteristics as well; that is, the current density threshold for high DOCPs decreases significantly. Figure 4b depicts the helicity-resolved EL spectra around the onset of the DOCP enhancement. Similar to the results for 1G-spin-LEDs shown in Figure 3b, the main EL emission band peaks at around 1.38 eV, which is below the *E*_g_ of GaAs. The secondary emission band due to band-to-band recombination is also noticeable in the photon region around and above 1.42 eV. At *J* = 7.0 A/cm^2^, the intensities of the σ^+^ and σ^−^ components are comparable. At *J* = 8.0 A/cm^2^, intensity of the component begins increasing. A further increase in *J* increases the σ^+^ component, whereas the σ^−^ component clearly decreases. Figure 4c shows the photon energy dependence of *P_CP_*. The enhancement in *P_CP_* begins around the spectral peak at 1.35 eV, then extends toward the band gap. Finally, *P_CP_* reaches 1.0 in the wide photon energy region at *J* = 10.0 A/cm^2^. Figure 4d shows the *J*-dependence of the integrated intensities of σ+ and σ− EL components and *P_CP_* obtained from 2G-spin-LEDs with different sizes. The data obtained from 1G-spin-LEDs have been included here for comparison. The separations of major (σ+) and minor (σ−) helicity EL components in 1G-spin-LEDs are observed at *J* region larger than approximately 8.0 A/cm^2^. Beyond the separations, the major helicity intensities increase compared to those corresponding to the reduced minor helicity component. This indicates that the minor components can be converted to major components. Accordingly, the DOCP steeply increases above the threshold current density of sub-10 A/cm^2^, whereas the 1G-spin-LEDs exhibit a rather gentle increase in *P_CP_* with *J* and require *J* ~ 100 A/cm^2^ to achieve a high *P_CP_*. In addition, there is almost no dependence on the device size in the 2G-spin-LEDs. Comparison of the *P_CP_* − *J* curves led to the inference that 1G-spin-LEDs exhibit leaky electronic characteristics, whereas 2G-spin-LEDs enable the efficient use of flowing current on emission with a small leakage. Furthermore, the reproducibility and stability of CPL emission with high DOCPs are improved in the 2G-spin-LEDs. Approximately 60% of the 2G devices exhibit high DOCP emissions, whereas only 5% of the 1G-spin-LEDs exhibit *P_CP_* enhancement. The stability of the high DOCP emission is also confirmed from the observation that the emission with *P_CP_*~0.38 continued for more than 2 days with a fixed voltage of 8 V and a current of ~41 mA, as shown in Figure 4e. Further systematic and multidimensional experiments on 2G-spin-LEDs have enabled the acquisition of experimental data that are not accessible using 1G-spin-LEDs, and discuss the mechanism underlying the steep increase in DOCP.

This review presents only the experimental results obtained in the early stage of the development of 2G-spin-LEDs, and the rationale for the aberrant DOCP enhancement based on the experimental evidence can be found elsewhere.

## 4. Controllability of Circular Polarization

High-speed switching and arbitrary control of polarization, including its sign and value, can expand the range of applicability of spin-LEDs as practical CPL sources. These functions should enable access to and sense digital information with high density and sensitivity. The DOCP magnitude of light emitted from a spin-LED is associated with the spin population of the injected carriers, and its sign is associated with the predominant spin direction, which depends on the magnetization direction; that is, light emitted in the magnetization direction is right-handed CPL, and vice versa. Therefore, the helicity can be switched by inverting the magnetization direction. However, magnetization reversal with the application of an external electromagnet waste space and electric power, which decreases the advantages of spin-LEDs. Controlling polarization with less action force is necessary for practical use. A spin-LED device with a pair of magnetic electrodes whose remnant state is anti-parallelly magnetized can control the polarization, including its sign and magnitude, by selecting the electrified electrodes and tuning the amount of current flowing through each electrode, respectively. Oestreich et al. [86] proposed this concept in the early stage of research on spin-photonic devices; however, not until Nishizawa et al. reported the electrical polarization switching on LT-spin-LEDs at low temperature [87] and RT [88] had experimental demonstration been reported. One of the factors in the success of LT-spin-LEDs is the feasibility of obtaining a pair of electrodes with an anti-parallel magnetization configuration without applying external fields, as mentioned below.

The structure of the tested spin-LED device consisted of a GaAs-based DH, a 1 nm thick x-AlO*_x_* layer, and a pair of Fe electrodes, as schematically shown in Figure 5a. The DH structure was the same as that described in the previous section. Here, in the active layer doped with 1 × 10^18^ cm^–3^ carbon, the radiative recombination time was estimated to be in the subnanosecond range at low temperature and ~10 ns at RT [78]. A helicity switching frequency of the order of few gigahertz was expected to be possible based on the recombination time. A pair of Fe electrodes were deposited on the x-AlO*_x_* layer by electron beam evaporation and fabricated by photolithography as two 40 μm stripes with a spatial separation of 250 μm. The two electrodes had different thicknesses, 100 and 30 nm, providing a sufficiently large difference in the switching fields owing to the shape anisotropy. The magnetization of a chip with a pair of electrodes as a function of the external magnetic field applied along the long side of the rectangular electrode showed a two-step behavior owing to the different switching fields. In the plateau region between the two switching fields, 10 Oe and 50 Oe, the magnetizations of the two electrodes pointed in the mutually reversed directions. The application of external magnetic fields in opposite directions facilitated the achievement of stable remnant magnetization. Specifically, a large positive external field of *H* = +5 kOe was applied to realize the parallel configuration; then, a negative field corresponding to that in the plateau region (−30 Oe in this case) was applied to reverse the magnetization of only one electrode. Finally, the external field was removed. The obtained anti-parallel magnetization was stable owing to a loop of magnetic force, as schematically illustrated by the dotted line in Figure 5a. The polarization-controllable (PC-) spin-LED devices prepared in this manner could emit CPL with approximately 10% polarization and a sign corresponding to the magnetization direction. Figure 5d shows the helicity-dependent EL spectra obtained at RT for continuous current injection into the 100 nm and 30 nm electrodes with current densities of 4.0 A/cm^2^ and 3.8 A/cm^2^ in the upper and lower panel, respectively. The shapes of the two spectra are almost identical at the EL peak position (1.43 eV), and the DOCPs are almost the same in magnitude but have the opposite signs: +0.10 and −0.12. Figure 5c shows an equivalent circuit of a PC-spin-LED. From the perspective of an electrical circuit, a spin-LED device can be considered a composite device consisting of a tunnel diode and a *p–n* diode arranged in mutually opposite directions. In Figure 5c, a PC-spin-LED is shown in the part that is surrounded by a dotted line, which includes two tunnel diodes and a common *p–n* diode. For analyzing the device performance during high-frequency operations, the tunnel diodes should be replaced by equivalent parallel circuits comprised of capacitors and resistors, as shown inside the dotted balloons, because of the equivalent capacitance of the tunnel diodes become dominant during such operations.

Electrical polarization switching was implemented by sending square current waves to the two electrodes using a two-channel current source. The phases of the square current waves differed from each other by half of a period. The current densities were fixed to obtain almost the same EL intensity. Figure 6a–c show the experimental results for *f* = 1 kHz at 5 K and RT, and *f* = 100 kHz at RT, respectively. In all measurements, a periodic inversion of the polarization is observed across the zero polarization position according to the frequency of the electrical signal, *f*. At low temperatures, the switching is very steep, whereas the steepness of the oscillations is reduced at RT. As *f* increases, the squareness decreases further. Thus, it can be concluded that alternate driving currents injected into the two electrodes with an antiparallel magnetization configuration can provide clear polarization switching up to RT with frequencies up to 100 kHz. However, there is a margin for improvement in the steep switching at RT and with a higher frequency. As shown in Figure 5c, the capacitance of the two tunnel diodes in a PC-spin-LED becomes dominant at high-frequency operations. Therefore, the steepness in the current-injection switching gradually decreases with increasing frequency; finally, the optical switching starts lagging behind the electrical switching. This degradation can be improved by arranging the external circuits and using electrical filters to eliminate the delay.

In addition, in PC-spin-LED devices, adjustment of the current flowing into each electrode enable arbitrary manipulation of the polarization. The current densities are used in the range in which the EL intensities, *I*_100_ and *I*_30_, vary linearly; however *P_CP_* exhibits almost no variation with respect to the injection current. By sending different amounts of current density within this range to each electrode synchronously, the emission intensity ratio, *I*_100_/(*I*_100_ + *I*_30_), can be changed while maintaining a constant total emission intensity. Figure 6d,e present the experimental results at 5 K and RT, respectively. The horizontal axis represents the emission intensity ratio, *I*_100_/(*I*_100_ + *I*_30_). The values of zero and one on the horizontal axis correspond to the cases in which the current flows only into the 100 and 30 nm Fe electrodes, respectively. The vertical axis shows the resulting *P_CP_* values calculated from the helicity-dependent EL spectra. *P_CP_* can be controlled by tuning the current sent to each electrode continuously from negative through zero to positive values, from −0.9. to +0.12 at 5K, and from −0.11 to +0.09 at RT. These results demonstrate two functions of spin-LEDs: All-electrical helicity switching and arbitrary polarization control.

## 5. Circularly Polarized Light Detection

CPL beams irradiated on a spin-LED device can excite spin-polarized carriers in a semiconductor. When the excited spin-up and spin-down electrons are transported to a magnetic electrode, they are subjected to different resistances, reflecting the difference between the densities of state of the spin-up and spin-down states in a ferromagnet at injection into the electrode. Stated differently, excited spin-polarized electrons generate different electromotive forces according to the relations between their spin directions and the magnetization direction. Therefore, the difference in the photoelectromotive force can provide information about the polarization state of light [89,90]. When these sequential processes can be efficiently conducted, the corresponding spin-LED devices can act as CPL detectors. Hereafter, a spin-LED device with a CPL detection function is called a spin-photodiode (spin-PD), which is another representative spin-photonic device. Spin-PDs can convert optical helicity signals through the spin state of the carriers into electrical signals with neither external optical polarizing elements nor optical delay modulators, which are essential in optical communication with polarization. 

Most studies on spin-PDs have dealt with VT devices in which the top surface of the device is irradiated by a light beam, usually through magnetic metal contacts [91,92,93,94,95,96,97,98]. The VT-spin-PD has a large light-receiving area; thus, the resulting signal is large. However, similar to the spin-LEDs, the VT devices also have two limitations related to the magnetization direction and magnetic circular dichroism (MCD). As described in Section 2, the magnetization direction is aligned with the optical axis by using magnetic materials exhibiting PMA or by applying a large external magnetic field vertically. The MCD component due to the magnetic contact is superimposed on the signals of interest, thereby obscuring the accuracy of the data [99]. On the contrary, LT-spin-PDs are advantageous because they avoid these issues as well as facilitate precise chip-to-chip alignment for direct device-to-device optical communications. Moreover, LT-devices with pairs of electrodes enable quantitative detection of the DOCP using the difference between the electromotive forces of the anti-parallel magnetized electrodes. However, studies on LT-spin-PD are scarce.

Ikeda et al. [100] investigated the helicity-dependent photocurrents in a device with the same structure as that of an LT-spin-LED, i.e., Fe/x-AlO*_x_*/DH. CPL beams were irradiated at right angles on the cleaved sidewall of the device, which was perpendicular to the stripe-shaped electrodes. The electrons were excited with spin polarization in the same direction as the remnant magnetization, then transferred diffusively toward the electrode, and subsequently injected into a ferromagnet through a tunneling barrier. Consequently, these electrons generated a photoelectromotive force depending on the spin state, as well as the polarization state of light. The CPL conversion efficiency (*F*) has been used as the figure of merit for spin-PDs; this parameter is defined as: *F* = Δ*I*/*I*, where Δ*I* is the difference in the photocurrent corresponding to the σ^+^ and σ^−^ components of the irradiated CPL, i.e., Δ*I* = *I*(σ^+^) − *I*(σ^−^); and *I* is the total photocurrent, i.e., *I* = [*I*(σ^+^) + *I*(σ^−^)]/2. The lower panel in Figure 7a shows the magnetic field dependence of Δ*I* superimposed on the magnetization behavior of the electrode. The behavior of Δ*I* follows the hysteresis magnetization curves with a slight difference in the coercive field, indicating that the spin-dependent electromotive force is attributable to CPL. Even in an LT-device, the detected signals may include the effects of MCD owing to internal reflection on magnetic materials, whose contribution is smaller than those for VT-spin-PDs, in which the light travels directly through the electrode. Considering the internal reflection at the Fe/GaAs interface, the MCD contribution can be estimated to be of the order of 10^−4^ in Δ*I*, and thus, it hardly contributes to the signals. This feature is one of the advantages of LT devices. CPL detection on spin-PDs having the same structure as those of spin-LEDs has been demonstrated from 5 K to 300 K; however, *F* is rather small (*F*~0.02%). One of the reasons for the low *F* has been inferred to be the photo-excited electrons that travel toward the electrode against a potential upward slope in the depletion layer where they experience large scattering. This process is schematically illustrated in the upper right part of Figure 7a. 

Instead of DH, Roca et al. [101] used bulk *p*-GaAs with a depletion region showing a downward slope. This material allows the photogenerated electrons to efficiently drift towards the electrode (called 2G-spin-PDs), as shown in the upper right corner of Figure 7b. In this case, *F* is improved to approximately 0.13%; however, the *F* − *H* profile does not show clear remanence in its hysteresis behavior (lower panel in Figure 7b). Another reason for the low *F* has been found to be a light-receiving volume that is concentrated near the cleaved edge at which the spin-injection junctions often suffer from degradation due to lateral irradiation and various external stimuli. To circumvent this issue, oblique-angle illumination was performed (Figure 7c). The *F* profile obtained closely matches that of the magnetization curve with hysteresis. A measurable *F* (i.e., *F* ~ 1.3%), which includes a contribution from the MCD effects, is observed at the remanence. To estimate the MCD contribution, the applied voltage dependences were investigated as the MCD component should be independent of the applied bias. Under a reverse bias, the Schottky depletion width increases and the built-in electric field becomes stronger. In this case, *F* remains nearly unchanged (*F* ~ 1.2%). In contrast, under a forward-bias, the Schottky depletion width is decreased, and the built-in field becomes weak. In this case, *F* is significantly decreased to *F* ~ 0.4%. This residual *F* is likely due to the MCD effect. Therefore, the net *F* is estimated to be approximately 0.9%. This value is approximately six times higher than that of the sidewall illumination and is comparable to that reported for a VT-spin-PD [92,95]; however, the MCD problem reemerges in this case. The calculations for 2G-spin-PDs reveal that the edge-related effects, such as magnetic edge curling [102] and defects in the x-AlO*_x_* tunnel barrier at the cleaved edge, are the origin of the experimentally measured low *F* for the sidewall illuminations. 

Therefore, Roca et al. [103,104] introduced a refracting-facet photodiode (RFPD) structure [105] into spin-PD devices (hereafter, referred to as 3G-spin-PDs), as shown in Figure 8a,b. In the RFPD structure, the light illuminated directly on the side of the device is bent by the refracting facet and sent directly onto an active layer just below a magnetic contact. The 3G-spin-PDs based on the RFPD structure are expected to circumvent the problems, such as absorption and transport near the spin-PD edges that are associated with cleaved edges. These problems lead to a poor helicity-dependent photocurrent. Furthermore, the negative DC bias applied between the magnetic electrode and back contact of the semiconductor is expected to increase the efficiency of transport of the photogenerated electrons toward the electrode. Figure 8c shows the bias voltage dependences of the photocurrent *I* and helicity-dependent components Δ*I*, and the inset depicts the behavior of *F*. *I* and Δ*I* increase with increasing reverse bias voltage, whereas a positive bias decreases them, as expected. Meanwhile, *F* remains nearly constant at approximately 0.4% with bias variation. This value is three times higher than that obtained from the edge-illuminated 2G-spin-PDs at RT, and thus, the MCD problem is avoided. In addition to experiments, a simulation model involving the optical selection rules, carrier and spin collection probability, and spin-dependent tunneling, has also been developed [106] based on the drift-diffusion equations for charge and spin [107,108,109] and spin-dependent tunneling equations [96,110,111]. The simulation results show that an *F* of up to 19% is achievable in 3G-spin-PDs with an Fe electrode. Further simulation-based analysis revealed that the discrepancy between the experimental and simulated *F* values is due to the low effective spin polarization of the Fe-based tunnel contact (~0.85%) near the interface, only if we consider that the observed discrepancy occurs solely due to the ferromagnetic contact. During the RFPD structure formation processes, the interface between the Fe layers and the x-AlO_x_ tunnel barrier undergoes degradation. This suggests that *F* can be significantly increased by improving the interface quality or by employing magnetic materials with higher polarization. 

Finally, we briefly discuss the effect that generates helicity-dependent photocarriers, namely the circular photogalvanic effect (CPGE). When a CPL is irradiated at an oblique incidence, a spin-polarized photocurrent flows orthogonally with respect to the plane of incidence of the irradiated light [112,113]. This phenomenon, called CPGE, is believed to specifically occur in low-dimensional structures wherein the Rashba-type spin-orbit interaction is dominant. Recently, a large CPGE, called anomalous CPGE, has been reported in a bulk GaAs with a vertically incident CPL; in this case, the spin-polarized photocurrent flows in all the radial directions, resulting in the generation of an electromotive force between both sides of the sample because of the inverse spin Hall effect [114]. Notably, the observed Δ*I/I* behavior possibly includes a sum of this anomalous CPGE and the magnetoresistance effect.

As described before, the functions of spin-photonic devices have been demonstrated individually to satisfy the requirements for actual use: (1) Compactness and integrability, (2) high DOCP emission (3) at RT (4) without applying an external magnetic field, (5) circular polarization controllability, and (6) CPL detection. However, there are numerous other issues that need to be addressed. Details on these issues are described in Section 7. 

## 6. CPL Applications with Spin-Photonic Devices

### 6.1. Proposed Applications Using CPL

As the next step in the development of spin-photonic devices, this section describes the CPL applications. Figure 9 illustrates the proposed applications using CPL. 

In communication, cryptographic techniques using CPL have been proposed [115,116]. In these techniques, the CPL can be utilized to carry the coded information. In the conventional telecommunication method, in which digital data are transformed into blinking lights, additional cryptographic information increases as the encryption becomes more complicated. In contrast, data encryption in the polarization state of light can be used as an efficient method for transferring the given data without significantly increasing the amount of information. Because circular polarizations, *σ*_+_ and *σ*_−_, as well as linear polarizations, are changed or lost by interception, eavesdropping can be detected. By applying this technique to wireless power feed systems in free space, only the designated devices can be charged. In short-haul communication between multiple fixed devices facing each other, the polarization information can be transferred without any loss. However, because CPL transmission through a curved optical fiber causes unpredictable depolarization, long-distance communication using CPL would require frequent relay systems to maintain its polarization. In addition, a compact CPL source is also expected to serve as a light source for writing information on magnetic recording media, as CPL can transfer angular momentum to the magnetic moments for precessional switching [117,118,119]. However, applications of CPL in telecommunications and recording techniques require the devices capable of emissions with high power and/or high coherency, that is, stimulated emission enhancement.

Stereoscopic displays based on right- and left-handed CPL have already been put into practice, such as in virtual reality for amusement attractions. These displays are realized by using two projectors with different polarization filters [120,121,122,123,124]. These views accommodate head tilt and prevent simulator sickness due to crosstalk between the helicities, which results from the rotational symmetry of CPL. As substitutes for the existing systems, integrated micro-CPL displays will provide significantly enhanced image resolution and reality. In addition, a circularly polarized ellipsometer has been proposed and studied [125,126]. The resulting periodic or oscillating emission of opposite helicities from integrated CPL emitters will be extremely effective in these applications.

The structural chirality in biomedical and chemical materials has a high adaptability to the optical chirality in CPL. Certain biological and polymer substances with some chirality exhibit different physiological and chemical activities associated with the polarity of CPL; these substances are called “enantiomers”. Some enantiomers have different effects in pharmaceuticals; for instance, one may be active, whereas the other may be non-active or noxious. CPL is often used for the identification and purification of enantiomers [127,128,129]. In addition, polarized light has been studied for the identification of heterogeneous biotissues [132]. When polarized light beams impinge upon a biological tissue, they are scattered multiple times by scatterers, mainly cell nuclei in the tissue. Depolarization of the resultant scattered light mainly depends on the size and axial ratio of the cell nuclei, as well as the frequency of scattering events associated with the density and distribution of the cell nuclei in the tissue. Few authors have proposed to use this technique to distinguish between closely related structural biological systems and observe the temporal structural changes. In particular, this technique is considered useful for identifying cancers in which cell nuclei become larger or distorted. The Mie scattering process is dominant when the scatterer is larger than the wavelength of the incident light [133]; this includes the case of scattering of visible and infrared light against cell nuclei. In the Mie regime, the degree of depolarization of CPL is much smaller than that of the linearly polarized light (LPL). In other words, the complete depolarization of CPL requires more scattering events than that of LPL [134,135]. Therefore, CPL scattering can provide more specific information about the outermost surfaces as well as the interior of tissues, which suggests the possibility of identifying carcinoma concealed deep in tissues.

Thus far, the proposed applications of CPL have been described. Spin-photonic devices could serve as key devices in these applications, although adaptation for each application is necessary. To realize CPL encryption technology and develop writing light sources for magnetic recording media, stimulated emission with a high DOCP and high switching frequency, comparable to the information processing speed, are necessary. To be employed in stereoscopic displays, polychromatic spin-LEDs in at least the three primary colors of light are necessary. The emission wavelengths of the spin-LEDs can change the bandgap energy of the active layer in the base LED semiconductor. Spin-LEDs capable of emitting shorter wavelengths (blue or ultraviolet) have been studied [36,37,38], although this research is still in progress. The achievement of laser emission or multiwavelength spin-LEDs requires further breakthroughs.

In contrast, spin-LEDs capable of spontaneous emission at near-infrared wavelengths could be utilized in biomedical identification without modifications. This is possible because the emission wavelengths of spin-LEDs are within the range from 650 to 1800 nm, the so-called “biological window,” in which the absorption due to both water and hemoglobin is small and scattering of light is minimal [136,137]. Moreover, optical coherency is not essential in this application. The introduction of spin-photonic devices into carcinoma identification techniques using CPL scattering would capable this technique to be developed from an *ex vivo* to an *in vivo* process. For example, spin-LEDs and spin-PDs integrated at the tip of a biopsy probe apparatus such as an endoscope, enable *in vivo* noninvasive cancer detection in real time while avoiding the unexpected risks associated with administering a fluorescent agent. Furthermore, CPL can provide information with depth resolution because it is robust against multiple scattering.

### 6.2. Cancer Identification Using CPL Scattering

Bickel et al. [132] reported that the polarization state of light differentially scattered from suspended biological scatterers can provide structural information about biological tissues. Since then, polarized light scattering has been considered useful for distinguishing between closely related structural biological systems and identifying subsequent time-dependent structural changes. Most of the earlier studies with the objective of observing biological structures using this technique were conducted using LPL [138,139,140], which successfully yielded surface information. However, little innovation has been achieved in bioimaging. Conversely, because there is a lack of CPL-based devices that are compatible with biological applications, the scattering technique with CPL remained unexplored for a long time. Recently, Meglinski et al. [130] pioneered the application of CPL in cancer detection by experimentally mapping the scattering properties of tissues on the Poincaré sphere. Kunnen et al. [131] reported that the light scattered from human lung tissue shows different polarization states for healthy and tumor tissues via *ex vivo* measurements using incident CPL (λ = 639 nm). They concluded that the resulting difference in polarization was caused by the enlargement of cell nuclei due to cancerization, and suggested that this technique could lead to the development of a noninvasive diagnostic technology for early disease detection. Subsequently, polarimetry with CPL and LPL has been widely studied to develop an optical diagnostic tool that can provide supplementary information to pathologists [141,142,143,144]. Moreover, polarimetry has been applied and demonstrated in the grading of colon cancer [145] and Alzheimer’s disease [146]. 

Nishizawa et al. [147] experimentally demonstrated cancer detection using CPL with a wavelength of approximately 900 nm which corresponds to the emission wavelength of spin-LEDs. Figure 10a shows the experimental setup used to measure the DOCP of the scattered light in various angular configurations. To demonstrate the cancer detection technique with this wavelength, the authors used CPL converted from an unpolarized laser beam (λ = 914 nm) using optical filters instead of an ideal spin-LED. This light was focused on the point of interest with an incident angle *θ*. The light scattered from the sample at an angle of *φ* ± 5° was collected and detected using a polarimeter with a high dynamic range (PAX1000; Thorlabs Inc.) as a substitute for an ideal spin-PD. The polarization state of the scattered light was evaluated in terms of the DOCP. Sliced tissue specimens of liver metastasis were prepared from a murine xenograft model of the human pancreatic cancer SUIT2 cells. Figure 10b shows a micrograph of the specimen, together with a schematic map that indicates the characteristics. The light blue area surrounded by a dotted line shows the metastatic (cancerous) parts concealed in the normal (healthy) tissue. When viewed under a microscope, the cell nuclei in the cancerous parts appear larger than those in the healthy part, with diameters of approximately 11 μm in the cancerous part and 6 μm in the healthy part. Line-scanning experiments were performed at 18 points along the line represented by a red arrow, crossing the boundary between the healthy and cancerous tissues multiple times. Figure 10c shows the line-scanning experimental results obtained using optical configurations with detection angles (*θ*,*φ*) of (*θ*,0) and (0,*φ*) in the upper and lower panels, respectively. Clear differences in the DOCP are observed depending on the state of the biological tissue, i.e., between cancerous and healthy tissue. Based on the angular dependence in terms of *θ* and *φ*, the appropriate angles for obtaining effective data are found to be *θ* ≤ 50° and (*θ* − *φ*) ≥ 30°. Within these ranges, the differences between the DOCPs obtained from cancerous and healthy parts are approximately 0.2, which can be sufficiently resolved using the present spin-PDs.

The CPL scattering process and depth resolution of the cancer identification technique with CPL scattering was validated by conducting Monte Carlo (MC) simulations [148,149]. The MC simulations were performed for pseudo-tissues, which are aqueous dispersions of particles with 5.9 and 11.0 μm diameters in the healthy and cancerous parts, respectively. These particle sizes correspond to the typical sizes of cell nuclei in healthy and cancerous cells. Figure 11a depicts the optical geometry. The pseudo tissues were irradiated by CPL beams at an incident angle, *θ* of 1° from a spin-LED. To eliminate the influence of light reflected at the incident point, the scattered light emitted from areas as far as 1 mm away from the incident point was collected and analyzed for every detection angle *φ* ± 5°. In the area corresponding to the pseudo-tissues in Figure 11a, the calculated light beam paths are depicted under the condition that the detection angle is 25 ± 5° and the photon number is 500,000. Figure 11b–d show the *φ* dependences of the DOCP, intensity, and sampling depth for healthy (blue) and cancerous (red) tissues. Here, the sampling depth is defined as the maximum depth reached by more than 30% of the detected light beams. The difference in DOCP between the two types of pseudo-tissues is almost constant at approximately 0.2 whereas the intensity of the detected light shows almost the same behavior, with a peak at approximately φ=30°. The effective angular range for obtaining sufficient intensity was approximately 0°–60°. These results indicate that cancerous tissues can be discriminated from bio-tissues within a wide angular range with constant sensitivity to the tissue state. As shown in Figure 11d, the sampling depth decreases monotonically with increasing *ϕ* in the effective angular range. This behavior indicates that the sampling depth can be tuned by modulating the detection angle.

Most carcinomas in the digestive system, such as gastric or esophageal cancer, emerge in the surface layer and progress into the deeper layers. Generally, when the carcinoma is found only in the mucosa, it can be treated by endoscopy, whereas advanced carcinoma in the submucosa or deeper can be treated surgically because the carcinoma may metastasize to the lymph nodes or other organs [150]. Accurate measurement of the cancer arrival depth without tissue ablation provides important information that aids in treatment decisions. Present endoscopic diagnosis techniques, such as narrow band imaging (NBI) [151], are capable of diagnosis (presence of cancer) and qualitative diagnosis (the distinction between tumors and non-tumors) with very high sensitivity, whereas very few direct measurement techniques can provide a quantitative depth profile of the carcinoma. Quantitative diagnosis is performed by a conjecture based on the surface morphology, which significantly depends on the skill and experience of the doctor. As shown in Figure 11d, cancer identification using CPL scattering provides a quantitative diagnostic tool to directly measure the cancer progression. In fact, MC simulations for pseudo-tissues consisting of two layers, a cancerous part on top and healthy tissue at the bottom, indicate that the DOCP of scattered light depends on the thickness of the cancer and detection angles, suggesting that the depth affected by cancer can be deduced by scanning the detection angle [149].

In addition, an endoscopic probe comprising spin-LEDs was designed based on the experimental and simulation results. Figure 12a shows a schematic cross-section of an optical device assembly chip that consists of one spin-LED for CPL irradiation, spin-PDs for CPL detection and a parabolic mirror for separation of scattered light according to the detection angle. Consider an ideal case in which the surface of the distal end of the endoscope faces an objective tissue, and incident CPL is irradiated from a spin-LED onto the tissue at an incident angle of 1°. The scattered light beams are separated according to the detection angle by reflection from a parabolic mirror whose focal point is as far as 1 mm away from the incident point, and then detected by each spin-PD. Even if the surface of the objective tissue does not directly face the endoscope tip, discrimination and depth measurements are possible because the difference in DOCP between healthy and cancerous tissues can be obtained over a wide angular range, as shown in Figure 10c, and the depth profile of the cancerous tissue can be estimated based on the results shown in Figure 11b–d. Moreover, by combining the various angular configurations with the distance and tilt measured via the reflections of the illuminated light, cancer detection for tissues with curved or wrinkled surface will become possible. This structure can be used to investigate tissue conditions with depth resolution. Furthermore, high-speed polarization switching of a PC-spin-LED can significantly improve the sensitivity of the DOCP to weakly scattered light through synchronous detection between a spin-LED and spin-PDs. Figure 12b presents a schematic illustration of *in vivo* cancer diagnosis using the designed spin-photonic device assembly attached to the tip of an endoscope.

## 7. Conclusions and Future Prospects

We reviewed state-of-the-art LT-spin-photonic devices with a focus on the development of practical CPL devices. The first half of this article demonstrated the functions of spin-photonic devices according to the requirements for actual use. The second half described various proposed applications of CPL devices and focused on biological diagnosis using CPL scattering. 

Compared with VT-optical devices, LT-spin-photonic devices have extremely small areas for light emission and reception but long propagation waveguides. These structural features are beneficial because the optical axis of the CPL is readily aligned inside the confined area with minimal dissipation [55] and various interactions occur between the carriers and photons throughout the waveguide. These characteristics are effective for meeting the requirements for practical use. This review listed six requirements for practical light components. The first includes compactness and integrability, which are intrinsically satisfied because spin-LEDs are based on semiconductors. The second requirement is that the device should be stand-alone and thus can operate without applying an external magnetic field or irradiating excitation light from another source. Aside from the use of electrode materials exhibiting PMA, adopting an LT-device structure aids in accomplishing these demands. The third requirement is RT operation, which is achieved by an improved tunneling barrier layer suitable for optical semiconductor devices. The fourth requirement is emission with a high DOCP. Surprisingly, almost fully polarized emission was observed from the LT-spin-LEDs. The fifth factor is polarization controllability. Spin-LED devices with a pair of electrodes with anti-parallel magnetization demonstrate high-speed reversal of the polarization sign by electrical switching between two electrodes as well as arbitrary polarization control by tuning the current ratio between the two electrodes. The sixth and final requirement is CPL detection, which has been demonstrated over a wide temperature range from 4 K to RT. 

Although these requirements are already satisfied, albeit minimally, further improvements are essential to realizing CPL devices based on the present spin-LED devices. The largest issue in device preparation is manufacturing yield improvement; in other words, reproducibility and stability of the spin-LEDs, exhibiting high CP emission and polarization controllability, are essential. Only a small percentage of the available spin-LEDs can stably emit almost pure CPL, with yields of approximately 5% for 1G-spin-LEDs and 60% for 2G-spin-LEDs. These yields are not close to the required commercial yield. Such low yields render it difficult to design and perform systematic experiments to understand the operation mechanism and optimize the device structure. This drawback may raise concerns about the credibility of the demonstration of pure CPL emission. Such low yields are mostly caused by the chemical instability of the ultrathin oxide layer, which is the key component in any spin-photonic device. This film is sensitive to external stimuli, such as heating above 200 °C, a strongly alkaline solution in conventional lithography, and ultrasonic vibration in the wire-bonding process. The damaged oxide layers caused by these stimuli reduce the resistance, resulting in a low spin-injection efficiency, low breakdown voltage, and large leakage current. In particular, these harmful effects are remarkable in spin-PDs with side illumination. In conventional microprocesses for a semiconductor device, annealing (sintering) is conducted to form a metal on the semiconductor in the alloy and reduce the contact resistance. The device isolation, for which silicon oxide is conventionally used, requires deposition at temperatures higher than 250 °C in some cases. Moreover, high-temperature annealing is employed to stabilize the magnetism of the magnetic electrodes. Moreover, high-temperature processes promote the formation of undesirable materials composed of semiconductors, oxides and metals at the interfaces in spin-LEDs. In particular, GaAs-based devices enhance the formation of lattice vacancies due to the desorption of V-group elements in the semiconductor. These vacancies act as nonradiative recombination centers. Moreover, although aluminum oxide, a type of ceramic, has a low solution velocity against the alkaline solution, which is used as a developer in lithography, ultrathin AlO*_x_* films (~ 1 nm) suffer fatal damage even with short exposure times and thus can often be locally removed. Furthermore, this film can deteriorate owing to the high temperature of the soldering iron and ultrasonic vibrations from the wire-welding equipment during the wiring processes. Therefore, microprocesses that are more suitable for unstable oxide films are required. Such processes should not involve high-temperature annealing, contact of an oxide film with a developer, and in-plane arrangement of the electrode pad with welding points far from the spin-injection points. The development of microprocesses will lead to improved reproducibility of spin-LEDs capable of emission with high DOCPs, which will eventually assist in assessing the operation mechanism and facilitating optimization of the device structure. Consequently, the device quality will drastically increase.

In the case of PC-spin-LEDs, switching and arbitrary control of CPL with a high DOCP is required. Currently, these functions have been demonstrated only within the DOCP range of ±0.15 at most. The above-mentioned device optimization will improve the DOCP range in which these functions are achievable. Therefore, this issue is subordinate to those previously mentioned. Spin-LEDs with two electrodes experience an additional issue concerning the emission point. In an LT-spin-LED, the CPL with the highest intensity and polarization is emitted from a point in the side facet just under the electrode. Both the intensity and polarization decrease further from the highest point. In a spin-LED with the two electrodes, the intensity and polarization are horizontally distributed with two strongest points just under the two electrodes. The switching and arbitrary control functions are beneficial for the total emission; however, the helicity switching and control would be asymmetrical when viewed locally. To increase the accuracy of DOCP control, apertures are attached to the facet during the device miniaturization to restrict the emission point. 

In the case of spin-PDs, the issues that required improvement are roughly divided into two types. The first issue concerns the improvement of the conversion efficiency *F.* A model simulation predicted that *F* could be increased to 19% when using an Fe electrode, and it could be increased up to 30% by replacing the ferromagnetic materials with Co_2_FeMnSi ($P_{FM}>0.8$ around $E_{F}$) However, *F* is drastically decreased due to extremely low polarization near the semiconductor interface, although the bulk exhibits high polarization. This characteristic presumably results from undesired chemical reactions, such as additional oxidation, electromigration, or diffusion of constituent atoms toward adjacent layers, which are caused by chemical stimuli during the preparation as well as by the illumination of a highly intense light and exposure to air. In 1G- and 2G- spin-PDs, these reactions are vigorous particularly near the cleaved edge, requiring protection of the edge structure. As another solution, a method of optically restricting the area in which light excites photocurrent with no polarimetric information could be proposed to increase *F*. Photocurrents with profitable information can be excited at the active layer just under the electrode, whereas the photocurrents with ambiguity increase according to the horizontal distance from the area just under the electrode. Therefore, a slit-like aperture with a rectangular window on the facet under the stripe electrode will decrease the superfluous photocurrents, thereby increasing *F*. An interdigital (comb) electrode will also decrease the redundant photocurrents. Another issue concerning the spin-PDs is quantitative CPL detection on a spin-PD device with two electrodes. This function of spin-PDs is useful in various applications. Ferromagnetic electrodes utilized in either or both of the electrodes will provide information about the polarization of the impinging light through a difference or ratio of the electromotive force between them. In the case of a spin-PD with one ferromagnetic electrode and one normal metal electrode, the optically excited spins that are parallel to the magnetization direction flow into the ferromagnetic electrode; in contrast, the rest flow into the normal electrode, which is valid when the polarization sign of the impinging CPL is known. Conversely, similar to the PC-spin-LEDs, a pair of magnetic electrodes are magnetized anti-parallelly, enabling quantification of optical polarization through the difference in the electromotive force between them. These proposals should be demonstrated in future studies. 

Extensive efforts will also be necessary to overcome these challenging issues to develop the prototype spin-photonic devices described in this review into realistic CPL devices. Furthermore, with the extension of the research on LT-spin-LEDs, it may be possible to develop a CPL laser that can emit coherent CPL emission, a so-called spin laser. In light of the intense work and contributions from various fields, including physics, chemistry, electrical engineering, information science, artificial intelligence, and mathematical information theory, there is no doubt that the realization of spin-lasers as well as the proposed CPL applications will be achieved in time.

## Figures and Tables

**Figure 1 micromachines-12-00644-f001:**
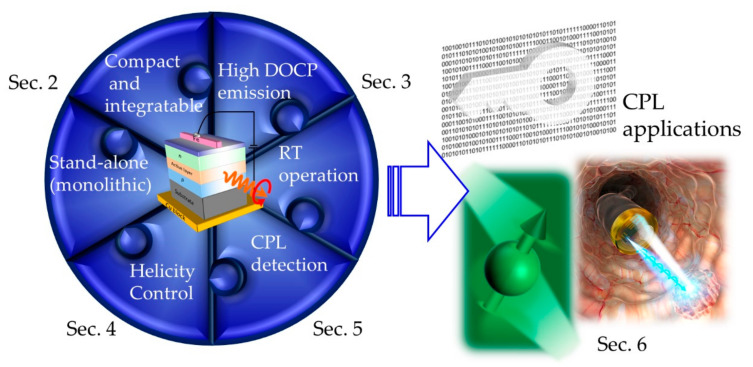
Schematic outline of this review with the corresponding section number. (**Left**) Requirements for practical CPL use with LT-spin-LED at the center. (**Right**) CPL applications: Quantum cryptography, holography, and biosensing using CPL, counterclockwise.

**Figure 2 micromachines-12-00644-f002:**
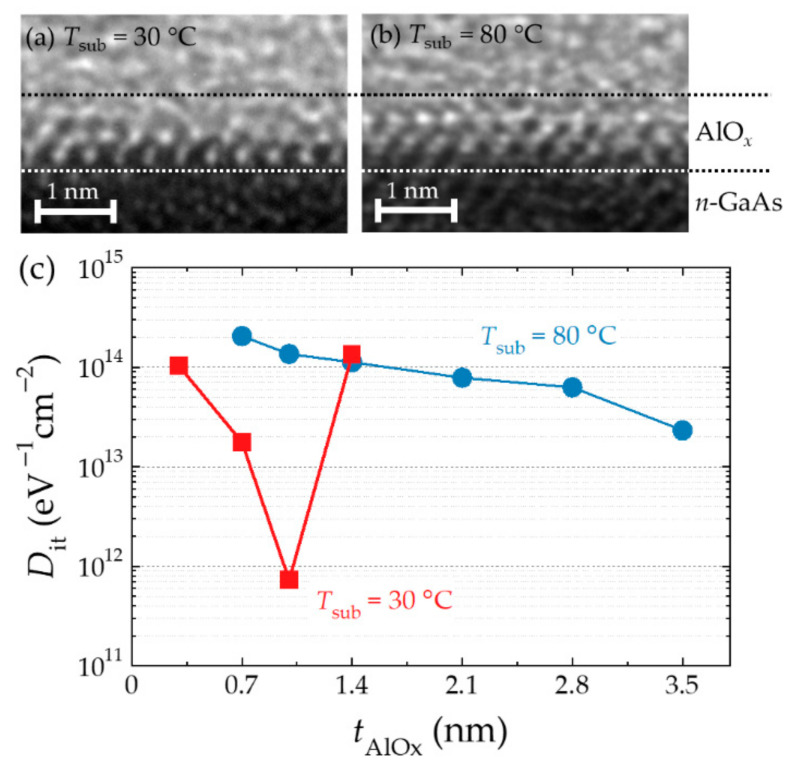
Cross-sectional TEM images of 1-nm-thick AlO*_x_* on *n*-GaAs with epitaxial Al layers grown at *T*_sub_ of (**a**) 30 °C and (**b**) 80 °C. The horizontal dotted lines roughly indicate the boundaries. (**c**) Thickness dependence of *D*_it_ for AlO*_x_* layers formed via post-oxidation of Al epilayer at *T*_sub_ = 30 °C (red) and 80 °C (black).

**Figure 3 micromachines-12-00644-f003:**
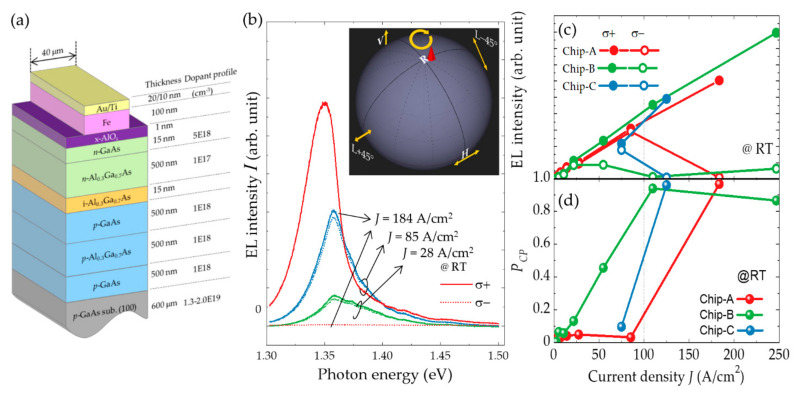
(**a**) Schematic structure of spin-LED for high DOCP with the layer thicknesses and doping profiles. (**b**) Helicity-dependent EL spectra at RT corresponding to three current densities: *J* = 28 (dotted curve), 85 (dashed curve), and 184 (solid curve) A/cm^2^. The red and blue curves show the σ+ and σ− EL components, respectively. Inset depicts the polarization state of the data with *J* = 184 A/cm^2^ on Poincaré sphere measured by a calibrated polarimeter. Plots of the (**c**) integrated intensity of σ+ (closed symbols) and σ− (open symbols) EL components and (**d**) *C_PC_* as a function of *J* for three different spin-LED chips. Reproduced with permission from [75].

**Figure 4 micromachines-12-00644-f004:**
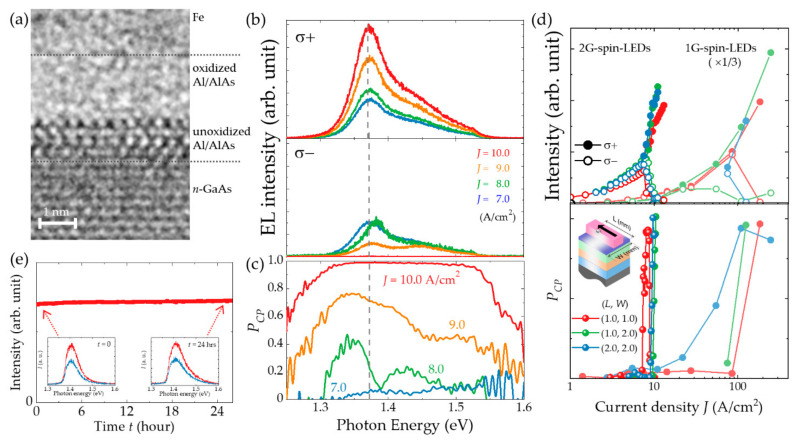
(**a**) Cross-sectional TEM image around the Fe/(oxidized Al/AlAs)/GaAs region in a spin-LED structure. The horizontal dotted lines roughly indicate the boundaries. (**b**) Helicity-resolved EL spectra at 7.0 (blue), 8.0 (green), 9.0 (orange), and 10.0 (red) A/cm^2^ obtained from 2G-spin-LED chips at RT. The upper and lower panels represent the integrated intensities of the σ+ and σ− EL components, respectively. The horizontal axis is common to (c). The vertical dashed line lies at 1.382 eV which is the peak of the spectra. (**c**) *C_PC_* spectra obtained from the data shown in (b). (**d**) Current density dependence of (upper panel) the integrated intensities of σ+ (closed symbols) and σ− (open symbols) EL components, and (lower panel) *C_PC_* obtained from several 2G-spin-LEDs chips with different sizes. The corresponding values for a 1G-spin-LED have been provided for comparison. The size parameters (*L*, *W*) are shown in the lower panel with the schematic definitions. (**e**) Results of lifetime test for a 2G-spin-LED at RT. The main panel depicts temporal net intensity profile measured without optical filters, whereas the insets show helicity dependent EL spectra (left) before the test and (right) after 24 h emission. Reproduced with permission from [82].

**Figure 5 micromachines-12-00644-f005:**
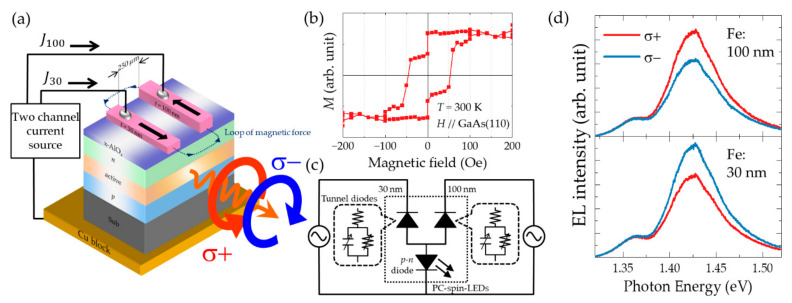
(**a**) Schematic illustration of a polarization-controllable (PC-) spin-LED together with a current source. (**b**) Magnetization curves for a chip with a pair of electrodes at RT. (**c**) Equivalent circuit of a PC-spin-LED. The part surrounded by the dotted line represents a PC-spin-LED, which consists of two tunnel diodes and a common *p–n* diode. The polarity of the tunnel diodes opposite to that of the *p–n* diode. The tunnel diodes can be replaced by the equivalent circuits shown in the balloons. (**d**) Helicity-specific EL spectra were obtained at RT by sending a continuous current to electrodes with thicknesses of (upper) 100 nm and (lower) 30 nm. Reproduced with permission from [87,88].

**Figure 6 micromachines-12-00644-f006:**
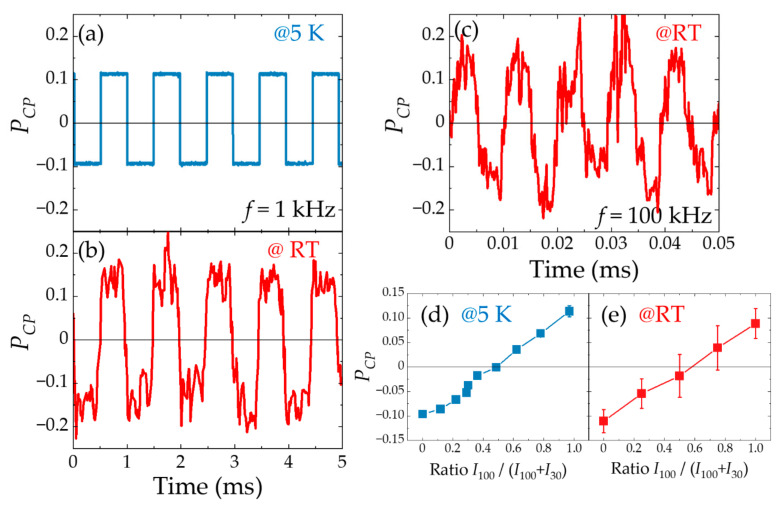
Experimental data for electrical polarization switching by sending square current waves into two electrodes with frequency *f* = 1 kHz at (**a**) 5 K and (**b**) RT, and (**c**) *f* = 100 kHz at RT. (**d**)**,** (**e**) Experimental results for arbitrary polarization control at (**d**) 5 K and (**e**) RT. The horizontal axis shows the EL intensity ratio *I*_100_/(*I*_100_ + *I*_30_). Reproduced with permission from [87,88].

**Figure 7 micromachines-12-00644-f007:**
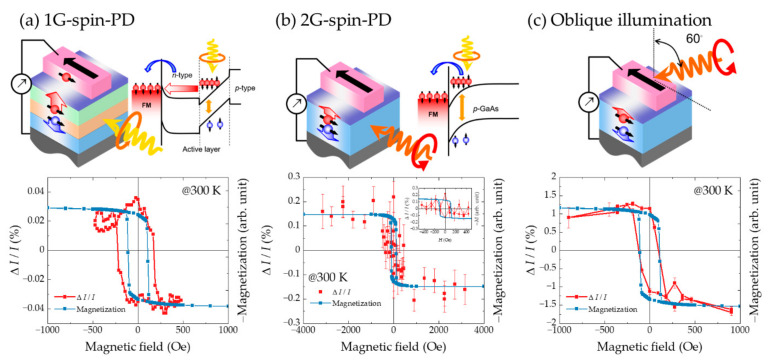
Schematic illustration of the device and experimental results of CPL detection for (**a**) a 1G-spin-PD with side illumination [100] and 2G-spin-PDs with (**b**) side illumination and **(c)** oblique illumination at 60° [101]. (**a**) and (**b**) also include schematic band diagrams in the upper right. The lower panels show the magnetic field dependence of the obtained photocurrent at RT and the electrode magnetization at 300 K. Reproduced with permission from [100,101].

**Figure 8 micromachines-12-00644-f008:**
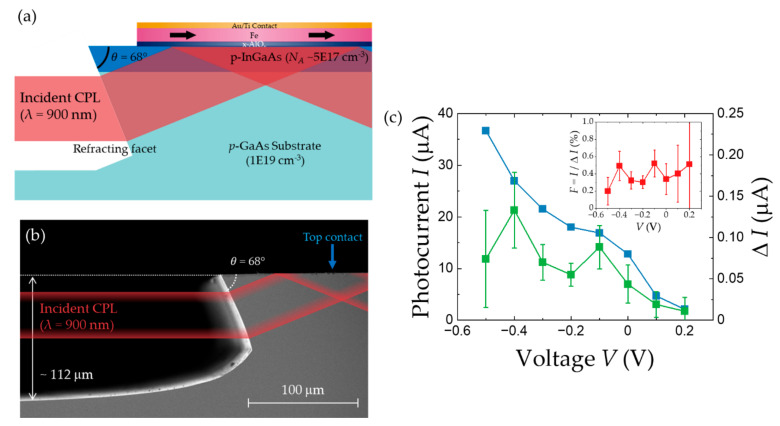
(**a**) Schematic cross–section of a 3G-spin-PD with a refracting facet. The magnetic electrode is located on the mesa structure, and the refracting facet is fabricated outside the stripe-shaped electrode via wet etching. The magnetization direction represented by the arrows in the Fe layer is parallel to the optical axis. A light beam is irradiated horizontally from the left on the refracting facet, bent by the facet, and sent onto the InGaAs active layer. (**b**) Cross-sectional view of the cleaved edge of a fabricated 3G-spin-PD observed by a scanning electron microscope. The facet angle *θ* is 68°. The facet height and etch depth are approximately 70 and 110 μm, respectively. (**c**) Experimental results for a 3G-spin-PD. Plots of the obtained (blue) photocurrent, (green) helicity-dependent photocurrent (Δ*I*), and (inset) as functions of the applied voltage. Reproduced with permission from [104].

**Figure 9 micromachines-12-00644-f009:**
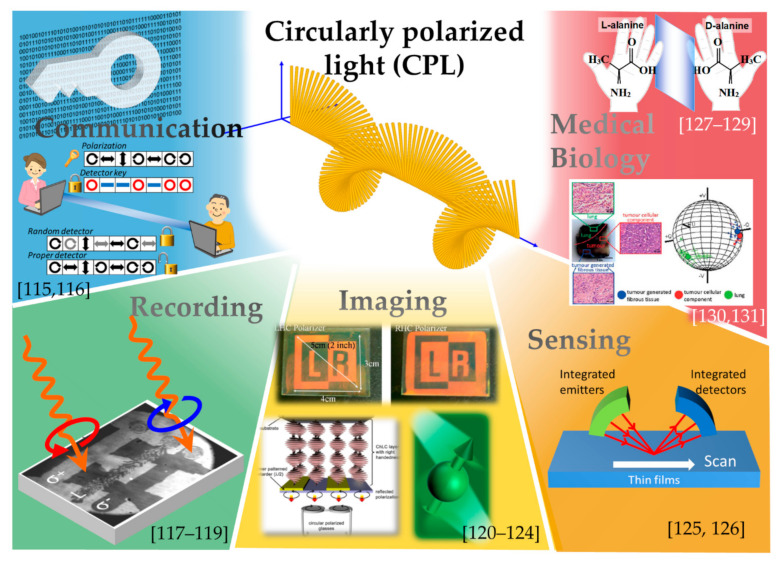
Proposed applications using CPL: (Counterclockwise from upper left) Cryptography in communication [115,116], light source for writing on magnetic media [117,118,119], 3D display and holography in imaging [120,121,122,123,124], scanning ellipsometry with CPL [125,126], and optical isomer separation and cancer diagnosis in medicine and biology [127,128,129,130,131]. Reproduced with permission from [118] and [131].

**Figure 10 micromachines-12-00644-f010:**
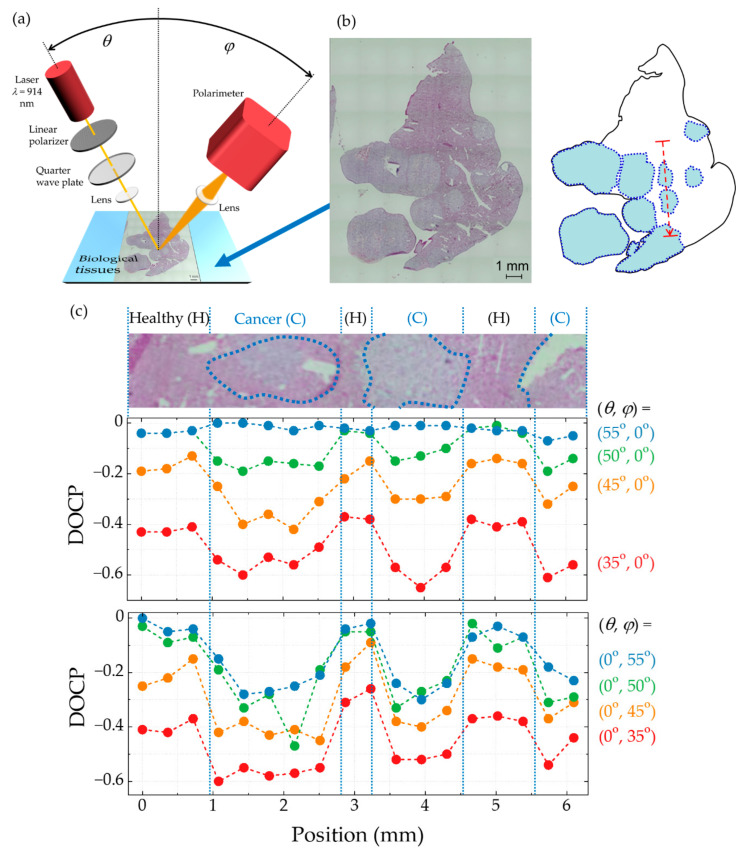
(**a**) Schematic illustration of the experimental setup for CPL illumination and measurement of DOCPs of the scattered light with various angular configurations. (**b**) (left) a micrograph of specimen and (right) a corresponding schematic map. The light blue areas delineated by the blue dotted lines represent the cancerous parts. The red arrow shows the area across healthy and cancerous parts in which line-scanning experiments were performed. **(c)** Results of the line-scanning experiments with different (upper) incident angles *θ* with *ϕ* = 0° and detection angles *ϕ* with *θ* = 0° along the red arrow shown in (**b**). The micrograph of the scanning area in the upper part corresponds to the probing points. Reproduced with permission from [147].

**Figure 11 micromachines-12-00644-f011:**
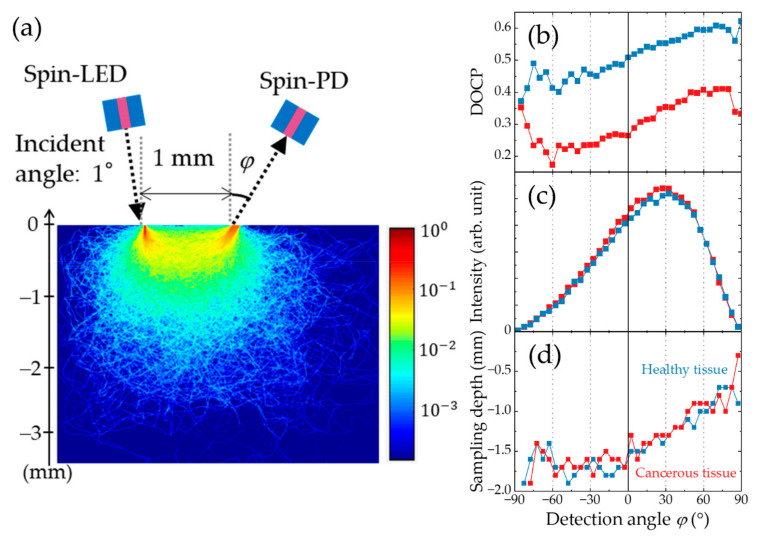
(**a**) Monte Carlo simulation geometry for multiple scattering in pseudo biological tissues, together with the calculated distribution of simulated light beam paths under the condition that *ϕ* = 25 ± 5° and the number of photons is 500,000. *ϕ* dependence of (**b**) DOCP, (**c**) intensity, and (**d**) sampling depth for pseudo-healthy tissues (*a* = 5.9 μm: blue plots and lines) and pseudo-cancerous tissues (*a* = 11.0 μm: red plots and lines). Reproduced with permission from [148,149].

**Figure 12 micromachines-12-00644-f012:**
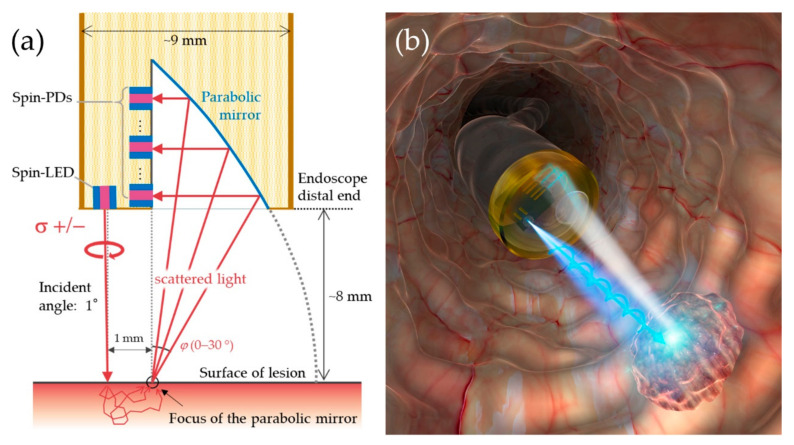
(**a**) The schematic cross-sectional design of an endoscope probe structure consisting of one spin-LED, an array of spin-PDs, and a parabolic mirror in the ideal case that the focus of the mirror is at the surface of the lesion. The scattered light beams from the detection point (the focal point of the mirror) with different angles are reflected by the mirror and detected by respective spin-PDs. Reproduced with permission from [148,149]. (**b**) Schematic illustration of *in vivo* cancer diagnosis with the designed CPL device assembly attached to the tip of an endoscope. Reproduced with permission from [147].

## Data Availability

The data that support the findings of this study are available from the corresponding author upon reasonable request.

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
