# Peer review of "Lateral-Type Spin-Photonics Devices: Development and Applications"

_micromachines, 2021, doi:10.3390/mi12060644_

Round 1

Reviewer 1 Report

This paper is a review of lateral-type spin LEDs and detectors and suggestion of these devices. Introduction is more or less reasonable, and suggestions of applications are interesting. But about pure CPL emission, no matter what you think, the data is unnatural. In the first place, the spin polarization of the side-emitting spin-LED has been not trusted. Who would believe in the polarization enhancement among people who know spin-LEDs well, even though published on the PNAS? Artifacts seemed to be removed in PNAS, but they were not perfect. Moreover, the origin has not been identified even though it has been more than 3 years. I think it is a faith issue as a researcher; if the authors want to write it, I don’t interfere. Therefore, I would request one minor revision. In page 2 line 61, ‘In the earliest studies on spin-LEDs [4, 5, 9-11], …’, the sample of ref. 4 is not a ferromagnetic semiconductor but paramagnetic one. So, please remove ref. 4 in these citations here.

Reviewer 2 Report

The authors have presented a review of the state-of-art of the spin LED and spin photodiode with a lateral type structure (light emitting and detection from the edge of device). They have also discussed in details of different requirements for real applications. In the end, they have given an example of application in biological diagnosis using CPL scattering. In global, this review paper is well organized and written. The paper allows the readers to understand the development of spin LED and spin photodiode during the two decades and helps the community to develop spin-photonic devices for different applications. However, some issues should be improved before the paper can be accepted for publication.

  1. In Page 3, line 106, the authors have cited the work of Truong et al [31] for the representative work of CoFeB/MgO spin injector with 15% CPL at RT. This is not an appropriate reference since the measurement was performed at 15K with a specific pulse current injection condition. Lu et al [Appl.Phys.Lett. 93, 152102 (2008)] has already published the work in 2008 with CoFeB/MgO in-plane magnetized spin injector, which shows a CPL up to 32% at 100K. This work should be cited as the first work for spin LED with CoFeB/MgO spin injector.
  2. In page 3, line 141, please add reference for the electrical controllability of the polarization for monolayer of TMDs.
  3. In page 5, line 187, please provide a number for the current-spreading width in a typical spin LED work condition.
  4. In page 6, line 263, there is an error on the unit, 7nm should be 7Å.
  5. In page 7, line 301, the sentence “ a 500 nm thick n-AlGaAs clad layer was adopted, which ensured 60% spin polarization…” is not clear. As the authors have mentioned in line 287 in the same page that 63% is the spin injection efficiency but not spin polarization, I don’t understand what is the 60% spin polarization here?
  6. In Page 9, Fig4, please add one figure for the comparison of EL intensity between 1G-spin LED and 2G-spin LED as a function of injection current density.
  7. In Page 9, Fig4d, please add the size of chips corresponding to the curves with different colors of 2G-spin LEDs.
  8. In Page 10, line 395-408, the authors have discussed the effect of oxidized Al/AlAs bilayer on the enhancement of spin injection efficiency. In fact, the effect of oxide barrier thickness on the efficiency of spin LED has also well studied in the Ref.[Appl.Phys.Lett. 93, 152102 (2008)], where the spin LED with thick barrier shows a much higher efficiency than the spin LED with thin barrier. The authors could also add this reference in line 406.
  9. In Page 12, line 508, the explanation of the decreasing of steep switching at RT and high frequency is not clear. The authors may add a schematic figure for the equivalent circuit and describe how to improve this circuit for future high speed operation.
  10. In page 14, Fig.7, the Y axis of the panel of magnetic field dependence of photocurrent should be ΔI/I (%). In (a) and (b), 1G-spin LED and 2G-spin LED should be 1G-spin PD and 2G-spin PD.
  11. In page 15, line 601, the F value with oblique illumination is ten times higher than that of sidewall illumination but the MCD reemerges. Can the authors give an estimation of the amplitude of MCD effect in this configuration?
  12. In page 16, line 630, the authors describe that the simulation show a high F value of 19%, but the experiment only find 0.4%. The discrepancy is due to the low effective spin polarization of Fe based tunnel contact of 0.85%. This explanation seems to be conflict with the demonstration of 63% spin injection efficiency with the same injector structure for spin LED measurement. The authors should improve their explanation.
  13. In page 20, line 798, the sampling depth should decrease with the increasing Ф.
  14. The idea for the application of cancer identification by using CPL scattering is very good. However, the presented Monte Carlo simulations or design of endoscope probe structure only work on a perfect flat surface with a rather small incident angle of 1°. The authors should discuss how to deal with the complicate situation in human body on a rolling surface.
  15. In page 23, line 931, Ec should be EF.
  16. The title and citation of Ref. [94] is not correct. The title is “Angular Dependence of the Spin Photocurrent in a Co-Fe-B/MgO/n-i-p GaAs Quantum-Well Structure”. The citation is Phys. Rev. Appl. 8, 064022 (2017).

Reviewer 3 Report

In this work, the authors have provided an in-depth review of current state-of-the-art lateral-type spin-photonic devices and their applications in the generation and detection of circularly polarized light (CPL). The manuscript is well-structured including a couple of milestone works. The authors focused on describing the device functionality in practical CPL applications from five different categories which are very informative. Particularly, the application in biological diagnosis sounds promising. Therefore, I would recommend the publication of this work if the following comments are addressed.

(1) Please consider adding a brief introduction about the fundamental principle of CPL emission/detection in the presented ferromagnet/GaAs-type heterostructures. Even though this work focuses on the device functionality, a clear physical picture of CPL generation and detection needs to be introduced as a review article.

(2) All the spin-LEDs and CPL detectors in this work mainly focused on the GaAs and their derivatives which is fine. Recently there is a new material family (i.e., hybrid organic-inorganic perovskites) which can also exhibit similar functionality. For example, Kim et al, Science 371 1129 (2021), Wang et al, Nature Communications 10, 129 (2019), etc.

(3) In Fig. 7 (c), the authors describe the CPL detection under the oblique illumination condition. Has the circularly photogalvanic effect (CPGE) been considered? CPGE effect will produce similar helicity-dependent photocurrent without the need for ferromagnetic contacts.

Round 2

Reviewer 2 Report

The authors have answered all questions and made corresponding modifications in the revised manuscript. The paper can be published now.

Reviewer 3 Report

The authors have addressed all the comments. Thus I recommend the publication of this work.